# Development of a Novel Anti-CD44 Variant 6 Monoclonal Antibody C_44_Mab-9 for Multiple Applications against Colorectal Carcinomas

**DOI:** 10.3390/ijms24044007

**Published:** 2023-02-16

**Authors:** Ryo Ejima, Hiroyuki Suzuki, Tomohiro Tanaka, Teizo Asano, Mika K. Kaneko, Yukinari Kato

**Affiliations:** 1Department of Molecular Pharmacology, Tohoku University Graduate School of Medicine, 2-1 Seiryo-machi, Aoba-ku, Sendai 980-8575, Japan; 2Department of Antibody Drug Development, Tohoku University Graduate School of Medicine, 2-1 Seiryo-machi, Aoba-ku, Sendai 980-8575, Japan

**Keywords:** CD44, CD44v6, monoclonal antibody, colorectal cancer

## Abstract

CD44 is a cell surface glycoprotein, and its isoforms are produced by the alternative splicing with the standard and variant exons. The CD44 variant exon-containing isoforms (CD44v) are overexpressed in carcinomas. CD44v6 is one of the CD44v, and its overexpression predicts poor prognosis in colorectal cancer (CRC) patients. CD44v6 plays critical roles in CRC adhesion, proliferation, stemness, invasiveness, and chemoresistance. Therefore, CD44v6 is a promising target for cancer diagnosis and therapy for CRC. In this study, we established anti-CD44 monoclonal antibodies (mAbs) by immunizing mice with CD44v3-10-overexpressed Chinese hamster ovary (CHO)-K1 cells. We then characterized them using enzyme-linked immunosorbent assay, flow cytometry, western blotting, and immunohistochemistry. One of the established clones (C_44_Mab-9; IgG_1_, kappa) reacted with a peptide of the variant 6-encoded region, indicating that C_44_Mab-9 recognizes CD44v6. Furthermore, C_44_Mab-9 reacted with CHO/CD44v3-10 cells or CRC cell lines (COLO201 and COLO205) by flow cytometry. The apparent dissociation constant (*K*_D_) of C_44_Mab-9 for CHO/CD44v3-10, COLO201, and COLO205 was 8.1 × 10^−9^ M, 1.7 × 10^−8^ M, and 2.3 × 10^−8^ M, respectively. C_44_Mab-9 detected the CD44v3-10 in western blotting, and partially stained the formalin-fixed paraffin-embedded CRC tissues in immunohistochemistry. Collectively, C_44_Mab-9 is useful for detecting CD44v6 in various applications.

## 1. Introduction

Colorectal cancer (CRC) has become the second cancer type for the estimated deaths in men and women combined in the United States, 2023 [1]. The development of CRC is classically explained by Fearon and Vogelstein model; the sequential genetic changes including *APC* (adenomatous polyposis coli), *KRAS*, *DCC* (deleted in colorectal cancer, chromosome 18q), and *TP53* lead to CRC progression [2]. However, CRC exhibits heterogeneous outcomes and drug responses. Therefore, the large-scale data analysis by an international consortium classified the CRC into four consensus molecular subtypes, including the microsatellite instability immune, the canonical, the metabolic, and the mesenchymal types [3]. In addition, various marker proteins have been investigated for the prediction of prognosis and drug responses of CRC [4,5]. Among them, recent studies suggest that CD44 plays a critical role in tumor progression through its cancer-initiating and metastasis-promoting properties [6].

CD44 is a polymorphic integral membrane protein, which binds to hyaluronic acid (HA), and contributes to cell-matrix adhesion, cell proliferation, migration, and tumor metastasis [7]. When the CD44 is transcribed, its pre-messenger RNA can be received alternative splicing and maturated into mRNAs that encode various CD44 isoforms [8]. The mRNA assembles with ten standard exons and the sixth variant exon encodes CD44v6, which plays critical roles in cell proliferation, migration, survival, and angiogenesis [9,10]. Functionally, CD44v6 can interact with HA via the standard exons-encoded region [11]. Furthermore, the v6-encoded region functions as a co-receptor of various receptors for epidermal growth factor, hepatocyte growth factor, C-X-C motif chemokine 12, and osteopontin [12]. Therefore, the receptor tyrosine kinase or G protein-coupled receptor signaling pathways are potentiated in the presence of CD44v6 [13]. These functions are essential for homeostasis or regeneration in normal tissues. Importantly, CD44v6 overexpression plays a critical role in CRC progression. For instance, CD44v6 is involved in colorectal carcinoma invasiveness, colonization, and metastasis [14]. Therefore, CD44v6 is a promising target for cancer diagnosis and therapy.

The clinical significance of CD44v6 in CRC deserves consideration. Anti-CD44v6 therapies mainly include the blocking of the v6-encoded region by monoclonal antibody (mAb) [12]. First, humanized anti-CD44v6 mAbs (BIWA-4 and BIWA-8) labeled with ^186^Re exhibited therapeutic efficacy in head and neck squamous cell carcinoma (SCC) xenograft-bearing mice [15]. Furthermore, the humanized anti-CD44v6 mAb, bivatuzumab-mertansine (anti-tubulin agent) conjugate, was evaluated in clinical trials [16]. However, the clinical trials were discontinued due to severe skin toxicity, including a case of lethal epidermal necrolysis [17]. The efficient accumulation of mertansine was most likely responsible for the high toxicity [17,18]. Therefore, the development of anti-CD44v6 mAbs with more potent and fewer side effects is desired.

We established the novel anti-CD44 mAbs, C_44_Mab-5 (IgG_1_, kappa) [19] and C_44_Mab-46 (IgG_1_, kappa) [20] by Cell-Based Immunization and Screening (CBIS) method and immunization of CD44v3-10 ectodomain, respectively. Both C_44_Mab-5 and C_44_Mab-46 recognize the first five standard exons-encoding sequences [21,22,23]. Therefore, they can recognize both CD44s and CD44v (pan-CD44). Furthermore, C_44_Mab-5 and C_44_Mab-46 exhibited high reactivity for flow cytometry and immunohistochemical analysis in oral [19] and esophageal [20] SCCs. C_44_Mab-5 reacted with oral cancer cells such as Ca9-22, HO-1-u-1, SAS, HSC-2, HSC-3, and HSC-4 using flow cytometry [19]. Moreover, immunohistochemical analysis revealed that C_44_Mab-5 detected 166/182 (91.2%) of oral cancers [19]. In contrast, C_44_Mab-46 reacted with esophageal squamous cell carcinoma (ESCC) cell lines (KYSE70 and KYSE770) using flow cytometry [20]. In immunohistochemical analyses using C_44_Mab−46 against ESCC tissue microarrays, C_44_Mab−46 stained 63 of 67 (94.0%) cases of ESCC [20].

We also examined the antitumor effects of C_44_Mab-5 in mouse xenograft models [24]. We converted the mouse IgG_1_ subclass antibody (C_44_Mab-5) into an IgG_2a_ subclass antibody (5-mG_2a_), and further produced a defucosylated version (5-mG_2a_-f) using FUT8-deficient ExpiCHO-S (BINDS-09) cells. In vitro analysis demonstrated that 5-mG_2a_-f showed moderate antibody-dependent cellular cytotoxicity (ADCC) and complement-dependent cytotoxicity activities against HSC-2 and SAS oral cancer cells. In vivo analysis revealed that 5-mG_2a_-f significantly reduced tumor growth in HSC-2 and SAS xenografts in comparison to control mouse IgG, even after injection seven days post-tumor inoculation. These results suggested that treatment with 5-mG_2a_-f may represent a useful therapy for patients with CD44-expressing oral cancers.

For epitope mapping of C_44_Mab-5, we employed the RIEDL tag system (“RIEDL” peptide and LpMab-7 mAb) [23]. We inserted the “RIEDL” peptide into the CD44 protein from the 21st to 41st amino acids. The transfectants produced were stained by LpMab-7 and C_44_Mab-5 in flow cytometry. C_44_Mab-5 did not react with the 30th-36th amino acids of the deletion mutant of CD44. Further, the reaction of C_44_Mab-5 to RIEDL tag-inserted CD44 from the 25th to 36th amino acids was lost, although LpMab-7 detected most of the RIEDL tag-inserted CD44 from the 21st to 41st amino acids. These results indicated that the epitope of C_44_Mab-5 for CD44 was determined to be the peptide from the 25th to 36th amino acids of CD44 using the RIEDL insertion for epitope mapping (REMAP) method.

In this study, we developed a novel anti-CD44v6 mAb, C_44_Mab-9 (IgG_1_, kappa) by CBIS method, and evaluated its applications, including flow cytometry, western blotting, and immunohistochemical analyses.

## 2. Results

### 2.1. Establishment of Anti-CD44v6 mAb, C_44_Mab-9

We employed the CBIS method to develop anti-CD44 mAbs. In the CBIS method, we prepared a stable transfectant as an immunogen. Then, we performed the high throughput hybridoma screening using flow cytometry (Figure 1). In this study, mice were immunized with CHO/CD44v3-10 cells. Hybridomas were seeded into 96-well plates, and CHO/CD44v3-10-positive and CHO-K1-negative wells were selected. After limiting dilution, anti-CD44 mAb-producing clones were finally established. We next performed an enzyme-linked immunosorbent assay (ELISA) to determine the epitope of each mAb. Among them, C_44_Mab-9 (IgG_1_, kappa) was shown to recognize the only CD44p351–370 peptide (EETATQKEQWFGNRWHEGYR), which is corresponding to variant 6-encoded sequence (Table 1).

### 2.2. Flow Cytometric Analysis of C_44_Mab-9 to CD44-Expressing Cells

We next confirmed the reactivity of C_44_Mab-9 against CHO/CD44v3-10 and CHO/CD44s cells by flow cytometry. As shown in Figure 2A, C_44_Mab-9 recognized CHO/CD44v3-10 cells in a dose-dependent manner, but neither CHO/CD44s (Figure 2B) nor CHO-K1 (Figure 2C) cells. The CHO/CD44s cells were recognized by a pan-CD44 mAb, C_44_Mab-46 [20] (Appendix A). Furthermore, C_44_Mab-9 also recognized endogenous CD44v6 in CRC cell lines as it reacted with both COLO201 (Figure 2D) and COLO205 (Figure 2E) in a dose-dependent manner.

Next, we determined the binding affinity of C_44_Mab-9 with CHO/CD44v3-10, COLO201, and COLO205 using flow cytometry. The dissociation constant (*K*_D_) of C_44_Mab-9 for CHO/CD44v3-10, COLO201, and COLO205 was 8.1 × 10^−9^ M, 1.7 × 10^−8^ M, and 2.3 × 10^−8^ M, respectively, indicating that C_44_Mab-9 possesses a moderate affinity for CD44v3-10 or endogenous CD44v6-expressing cells (Figure 3).

### 2.3. Western Blot Analysis

We next performed western blot analysis to assess the sensitivity of C_44_Mab-9. Total cell lysates of CHO-K1, CHO/CD44s, and CHO/CD44v3-10 were analyzed. As shown in Figure 4, C_44_Mab-9 detected CD44v3-10 as a more than 180-kDa band. However, C_44_Mab-9 did not detect any bands from lysates of CHO-K1 and CHO/CD44s cells. An anti-pan-CD44 mAb, C_44_Mab-46, recognized the lysates from both CHO/CD44s (~75 kDa) and CHO/CD44v3-10 (>180 kDa). These results indicated that C_44_Mab-9 specifically detects exogenous CD44v3-10.

### 2.4. Immunohistochemical Analysis Using C_44_Mab-9 against Tumor Tissues

We next examined whether C_44_Mab-9 could be used for immunohistochemical analyses using formalin-fixed paraffin-embedded (FFPE) sections. Since previous anti-CD44v6 mAbs could detect CD44v6 in SCC tissues at a high frequency, we first stained an oral SCC tissue. As shown in Figure 5A, C_44_Mab-9 exhibited clear membranous staining, and could clearly distinguish tumor cells from stromal tissues. In contrast, C_44_Mab-46 stained both (Figure 5B). We next investigated CRC sections. C_44_Mab-9 showed membranous staining in CRC cells, but not stromal tissues (Figure 5C). In contrast, C_44_Mab-46 also stained both (Figure 5D). These results indicated that C_44_Mab-9 is useful for immunohistochemical analysis of FFPE tumor sections.

## 3. Discussion

In this study, we developed C_44_Mab-9 using the CBIS method (Figure 1), and determined its epitope as variant 6 encoded region (Table 1). Then, we showed the usefulness of C_44_Mab-9 for multiple applications, including flow cytometry (Figure 2 and Figure 3), western blotting (Figure 4), and immunohistochemistry (Figure 5).

Anti-CD44v6 mAbs (clones 2F10, VFF4, VFF7, and VFF18) were previously developed, and mainly used for tumor diagnosis and therapy. The 2F10 was established by the immunization of CD44v3-10-Fc protein produced by COS1 cells. The exon specificity of the 2F10 was determined by indirect immunofluorescent staining of COS1 cells transfected with human CD44v cDNAs, including CD44v3-10, CD44v6-10, CD44v7-10, CD44v8-10, and CD44v10 [25]. Therefore, the 2F10 is thought to recognize the peptide or glycopeptide structure of CD44v6. However, the detailed binding epitope of 2F10 has not been determined.

The VFF series mAbs were established by the immunization of bacterial-expressed CD44v3-10 fused with glutathione *S*-transferase [26,27]. Afterward, VFF4 and VFF 7 were used in the immunohistochemical analysis [28], and VFF18 was humanized as BIWA-4 [15], and developed to bivatuzumab-mertansine drug conjugate for clinical trials [17,18]. The VFF18 bound only to the fusion proteins, containing a variant 6-encoded region. Furthermore, the VFF18 recognized several synthetic peptides, spanning the variant 6-encoded region in ELISA, and the WFGNRWHEGYR peptide was determined as the epitope [26]. As shown in Table 1, C_44_Mab-9 also recognized a synthetic peptide (CD44p351–370), which possesses the above sequence. In contrast, a synthetic peptide (CD44p361–380) possesses the FGNRWHEGYR sequence, which is not recognized by C_44_Mab-9. Therefore, C_44_Mab-9 and VFF18 recognize CD44v6 with a similar variant 6-encoded region. Detailed epitope mapping for C_44_Mab-9 is required in the future.

A mutated version of BIWA-4, called BIWA-8, was constructed for improving binding affinity. This was achieved by two amino acid mutations of the light chain without changing the humanized heavy chain [15]. The BIWA-8 was further engineered to chimeric antigen receptors (CARs). The CD44v6 CAR-T exhibited antitumor effects against primary human acute myeloid leukemia and multiple myeloma cells in immunocompromised mice [29]. Furthermore, the CD44v6 CAR-T also showed efficacy in xenograft models of lung and ovarian carcinomas [30], which is expected for a wider development toward solid tumors. However, Greco et al. demonstrated that the *N*-glycosylation of CD44v6 protects tumor cells from the CD44v6 CAR-T targeting [31]. This phenomenon is probably due to the masking of CD44v6 CAR binding by the *N*-glycosylation because the original VFF18 was established by bacterial-expressed CD44v3-10 immunization and recognized the peptidic epitope lacking the *N*-glycosylation [26]. In contrast, C_44_Mab-9 was established by immunization of CHO/CD44v3-10 cells, but recognizes a synthetic peptide (Table 1). Meanwhile, C_44_Mab-9 could detect more than 180 kDa, heavily glycosylated CD44v3-10 in western blot analysis (Figure 4). Further studies are required to reveal whether the *N*-glycosylation affects the recognition by C_44_Mab-9 for future application to CAR-T therapy.

The clinical significance of CD44v6 expression in patients with CRC using immunohistochemical analysis remains controversial. The elevated expression has been associated with poor prognosis, linked to adverse prognosis [32,33]. However, others have reported that CD44v6 expression is associated with a favorable outcome [34,35]. Various clones of anti-CD44v6 mAbs appeared to influence the outcome of the clinical significance. Among these clinical studies, Saito et al. used VFF18 and showed similar staining patterns of C_44_Mab-9 (Figure 5). They also found that CD44v6 expression was observed in poorly differentiated CRC without E-cadherin expression. Furthermore, the high CD44v6 expression exhibited a significant inverse correlation with E-cadherin expression and was found to be an independent poor prognostic factor in disease-free survival and overall survival [36]. In the future, we should evaluate the clinical significance of the C_44_Mab-9-positive CRC with E-cadherin expression.

Large-scale genomic analyses of CRCs defined 4 subtypes: (1) microsatellite instability immune; (2) canonical; (3) metabolic; (4) mesenchymal types [3]. Since the CD44v6 expression was observed in a part of CRC tissues (Figure 5), the relationship to the subtypes should be evaluated. In addition, the mechanism of CD44v6 upregulation including the transcriptional regulation and the v6 inclusion by alternative splicing should be determined. The inclusion of CD44 variant exons was reported to be promoted by the ERK-Sam68 axis [37]. Moreover, CD44v6 forms a ternary complex with MET and HGF, which is essential for the c-MET activation [38]. This positive feedback is a potential mechanism to promote the variant exon inclusion.

CD44v6-positive CRC cells exhibited cancer-initiating cell properties [39]. Cytokines, HGF, C-X-C motif chemokine 12, and osteopontin, secreted from tumor-associated fibroblasts, promote the CD44v6 expression in the cancer-initiating cells, which promotes migration and metastasis of CRC cells [14]. Clinically, circulating-tumor cells (CTCs), which express EpCAM, MET, and CD44, identify a subset with increased metastasis-initiating phenotype [40], suggesting that CD44v6 plays an important role in cancer-initiating cell property cooperating with MET. In addition, CTC culture methods, including two-dimensional (2D) expansion, 3D organoids/spheroids culture, and xenograft formation in mice, have been developed to evaluate the character of CTCs [41]. Therefore, the biological property to affect cell proliferation and invasiveness by C_44_Mab-9 should be investigated because CD44v6 can potentiate the MET signaling by forming the ternary complex with HGF [38]. Therefore, it would be valuable to examine the effect of C_44_Mab-9 on CTC proliferation in vitro and metastasis in vivo.

To evaluate the in vivo effect, we previously converted the IgG_1_ subclass of mAbs into a mouse IgG_2a_, and produced a defucosylated version. These defucosylated IgG_2a_ mAbs exhibited potent ADCC in vitro, and reduced tumor growth in mouse xenograft models [24,42,43,44,45,46,47,48]. Therefore, the production of a class-switched and defucosylated version of C_44_Mab-9 is required to evaluate the antitumor activity in vivo.

## 4. Materials and Methods

### 4.1. Cell Lines

Mouse multiple myeloma P3X63Ag8U.1 (P3U1) and CHO-K1 cell lines were obtained from the American Type Culture Collection (ATCC, Manassas, VA, USA). These cells were cultured in Roswell Park Memorial Institute (RPMI)-1640 medium (Nacalai Tesque, Inc., Kyoto, Japan), supplemented with 10% heat-inactivated fetal bovine serum (FBS; Thermo Fisher Scientific, Inc., Waltham, MA, USA), 100 U/mL penicillin, 100 μg/mL streptomycin, and 0.25 μg/mL amphotericin B (Nacalai Tesque, Inc.). Human colorectal cancer cell lines, COLO201 and COLO205, were obtained from ATCC and the Cell Resource Center for Biomedical Research Institute of Development, Aging, and Cancer at Tohoku University, respectively. The COLO201 and COLO205 were cultured in RPMI-1640 medium, supplemented with 10% heat-inactivated FBS, 100 units/mL of penicillin, and 100 μg/mL streptomycin (Nacalai Tesque, Inc.). All the cells were grown in a humidified incubator at 37 °C with 5% CO_2_.

CD44s cDNA was amplified using HotStar HiFidelity Polymerase Kit (Qiagen Inc., Hilden, Germany) using LN229 cDNA as a template. CD44v3-10 ORF was obtained from the RIKEN BRC through the National Bio-Resource Project of the MEXT, Japan. CD44s and CD44v3-10 cDNAs were subcloned into pCAG-Ble-ssPA16 vector possessing signal sequence and N-terminal PA16 tag (GLEGGVAMPGAEDDVV) [19,49,50,51,52], which is detected by NZ-1 [53,54,55,56,57,58,59,60,61,62,63,64,65,66,67,68]. CHO/CD44s and CHO/CD44v3-10 were established by transfecting pCAG-Ble/PA16-CD44s and pCAG-Ble/PA16-CD44v3-10 into CHO-K1 cells using a Neon transfection system (Thermo Fisher Scientific, Inc.).

### 4.2. Hybridoma Production

The female BALB/c mice (6-weeks old) were purchased from CLEA Japan (Tokyo, Japan). Animals were housed under specific pathogen-free conditions. All animal experiments were also conducted according to relevant guidelines and regulations to minimize animal suffering and distress in the laboratory. The Animal Care and Use Committee of Tohoku University (Permit number: 2019NiA-001) approved animal experiments. The mice were monitored daily for health during the full four-week duration of the experiment. A reduction of more than 25% of the total body weight was defined as a humane endpoint. During sacrifice, the mice were euthanized through cervical dislocation, after which death was verified through respiratory and cardiac arrest. The mice were intraperitoneally immunized with CHO/CD44v3-10 (1 × 10^8^ cells) and Imject Alum (Thermo Fisher Scientific Inc.) as an adjuvant, which stimulates a nonspecific immune response for mixed antigens using this formulation of aluminum hydroxide and magnesium hydroxide. After three additional immunizations of CHO/CD44v3-10 (1 × 10^8^ cells), a booster injection of CHO/CD44v3-10 was intraperitoneally administered 2 days before harvesting the spleen cells. The splenocytes were fused with P3U1 cells using polyethylene glycol 1500 (PEG1500; Roche Diagnostics, Indianapolis, IN, USA). The supernatants, which are positive for CHO/CD44v3–10 cells and negative for CHO-K1 cells, were selected by the flow cytometry-based high throughput screening using SA3800 Cell Analyzers (Sony Corp., Tokyo, Japan).

### 4.3. ELISA

Fifty-eight synthesized peptides (Sigma-Aldrich Corp., St. Louis, MO, USA), which cover the CD44v3-10 extracellular domain [21], were immobilized on Nunc Maxisorp 96-well immunoplates (Thermo Fisher Scientific Inc) at a concentration of 1 µg/mL for 30 min at 37 °C. After washing with phosphate-buffered saline (PBS) containing 0.05% (*v*/*v*) Tween 20 (PBST; Nacalai Tesque, Inc.) using Microplate Washer, HydroSpeed (Tecan, Zürich, Switzerland), wells were blocked with 1% (*w*/*v*) bovine serum albumin (BSA)-containing PBST for 30 min at 37 °C. C_44_Mab-9 was added to each well, and then incubated with peroxidase-conjugated anti-mouse immunoglobulins (1:2000 diluted; Agilent Technologies Inc., Santa Clara, CA, USA). Enzymatic reactions were performed using 1 Step Ultra TMB (Thermo Fisher Scientific Inc.). The optical density at 655 nm was measured using an iMark microplate reader (Bio-Rad Laboratories, Inc., Berkeley, CA, USA).

### 4.4. Flow Cytometry

CHO-K1 and CHO/CD44v3-10 were isolated using 0.25% trypsin and 1 mM ethylenediamine tetraacetic acid (EDTA; Nacalai Tesque, Inc.) treatment. COLO201 and COLO205 were isolated by brief pipetting. The cells were treated with primary mAbs or blocking buffer (0.1% bovine serum albumin (BSA; Nacalai Tesque, Inc.) in phosphate-buffered saline [PBS]; control) for 30 min at 4 °C. Subsequently, the cells were incubated in Alexa Fluor 488-conjugated anti-mouse IgG (1:2000; Cell Signaling Technology, Inc.) for 30 min at 4 °C. Fluorescence data were collected using the SA3800 Cell Analyzer and analyzed using SA3800 software ver. 2.05 (Sony Corporation).

### 4.5. Determination of Dissociation Constant (K_D_) by Flow Cytometry

Serially diluted C_44_Mab-9 was suspended with CHO/CD44v3-10, COLO201, and COLO205 cells. The cells were further treated with Alexa Fluor 488-conjugated anti-mouse IgG (1:200). Fluorescence data were collected using BD FACSLyric and analyzed using BD FACSuite software version 1.3 (BD Biosciences). To determine the dissociation constant (*K*_D_), GraphPad Prism 8 (the fitting binding isotherms to built-in one-site binding models; GraphPad Software, Inc., La Jolla, CA, USA) was used.

### 4.6. Western Blot Analysis

The cell lysates (10 μg of protein) were separated on 5–20% polyacrylamide gels (FUJIFILM Wako Pure Chemical Corporation, Osaka, Japan) and transferred onto polyvinylidene difluoride (PVDF) membranes (Merck KGaA, Darmstadt, Germany). After blocking (4% skim milk (Nacalai Tesque, Inc.) in PBS with 0.05% Tween 20), the membranes were incubated with 10 μg/mL of C_44_Mab-9, 10 μg/mL of C_44_Mab-46 or 1 μg/mL of an anti-β-actin mAb (clone AC-15; Sigma-Aldrich Corp.), and then incubated with peroxidase-conjugated anti-mouse immunoglobulins (diluted 1:1000; Agilent Technologies, Inc., Santa Clara, CA, USA). Finally, the signals were detected with a chemiluminescence reagent, ImmunoStar LD (FUJIFILM Wako Pure Chemical Corporation) using a Sayaca-Imager (DRC Co., Ltd., Tokyo, Japan).

### 4.7. Immunohistochemical Analysis

The FFPE oral SCC tissue was obtained from Tokyo Medical and Dental University [69]. FFPE sections of colorectal carcinoma tissue array (Catalog number: CO483a) were purchased from US Biomax Inc. (Rockville, MD, USA). The sections were autoclaved in citrate buffer (pH 6.0; Nichirei biosciences, Inc., Tokyo, Japan) for 20 min. After blocking with SuperBlock T20 (Thermo Fisher Scientific, Inc.), the sections were incubated with C_44_Mab-9 (1 μg/mL) and C_44_Mab-46 (1 μg/mL) for 1 h at room temperature and then treated with the EnVision+ Kit for mouse (Agilent Technologies, Inc.) for 30 min. The color was developed using 3,3′-diaminobenzidine tetrahydrochloride (DAB; Agilent Technologies Inc.) for 2 min. Hematoxylin (FUJIFILM Wako Pure Chemical Corporation) was used for the counterstaining. Leica DMD108 (Leica Microsystems GmbH, Wetzlar, Germany) was used to examine the sections and obtain images.

## Figures and Tables

**Figure 1 ijms-24-04007-f001:**
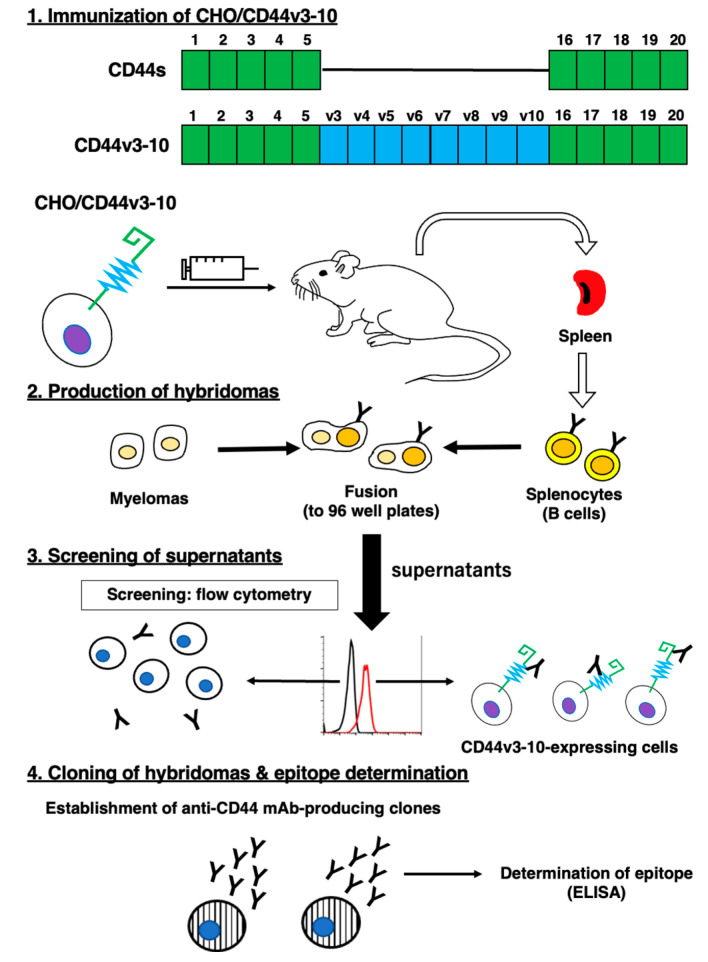
A schematic illustration of ant-human CD44 mAbs production. A BALB/c mouse was intraperitoneally immunized with CHO/CD44v3-10 cells. Hybridomas were produced by the fusion of the splenocytes and P3U1 cells. Then, the screening was performed by flow cytometry using parental CHO-K1 and CHO/CD44v3-10 cells. After cloning and additional screening, a clone C_44_Mab-9 (IgG_1_, kappa) was established. Finally, the binding epitopes were determined by enzyme-linked immunosorbent assay (ELISA) using peptides that cover the extracellular domain of CD44v3-10.

**Figure 2 ijms-24-04007-f002:**
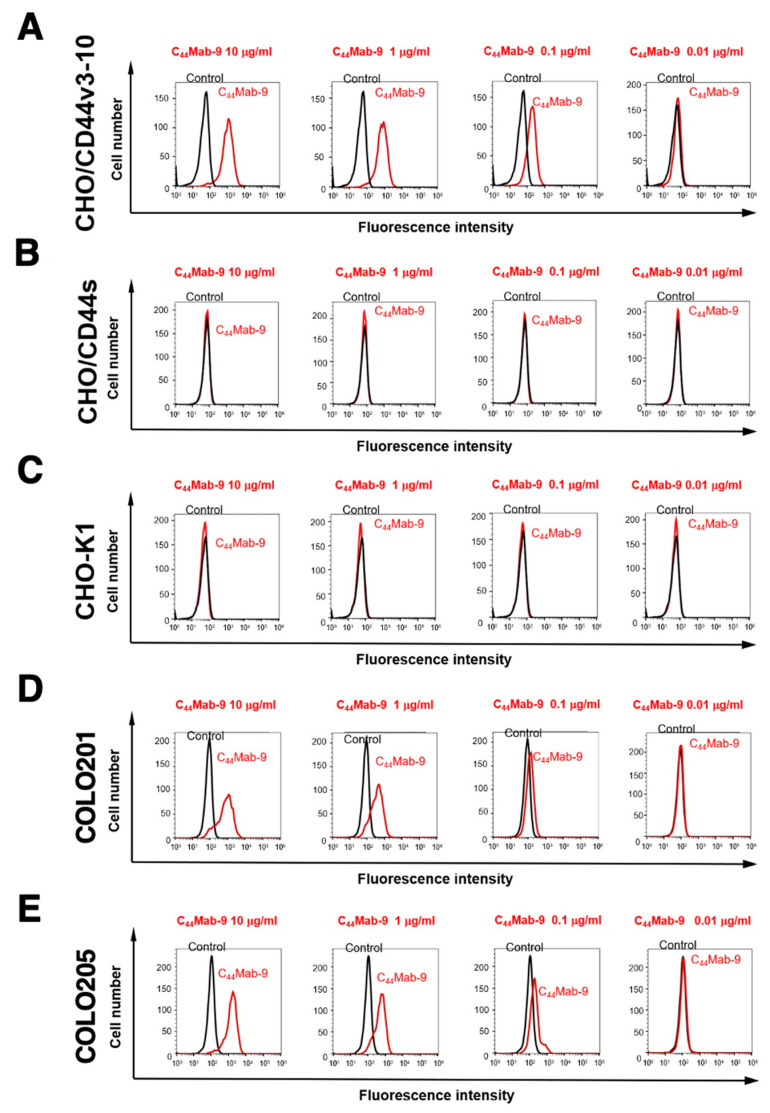
Flow cytometry to CD44-expressing cells using C_44_Mab-9. CHO/CD44v3-10 (**A**), CHO/CD44s (**B**), CHO-K1 (**C**), COLO201 (**D**), and COLO205 (**E**) were treated with 0.01–10 µg/mL of C_44_Mab-9, followed by treatment with Alexa Fluor 488-conjugated anti-mouse IgG (Red line). The black line represents the negative control (blocking buffer).

**Figure 3 ijms-24-04007-f003:**
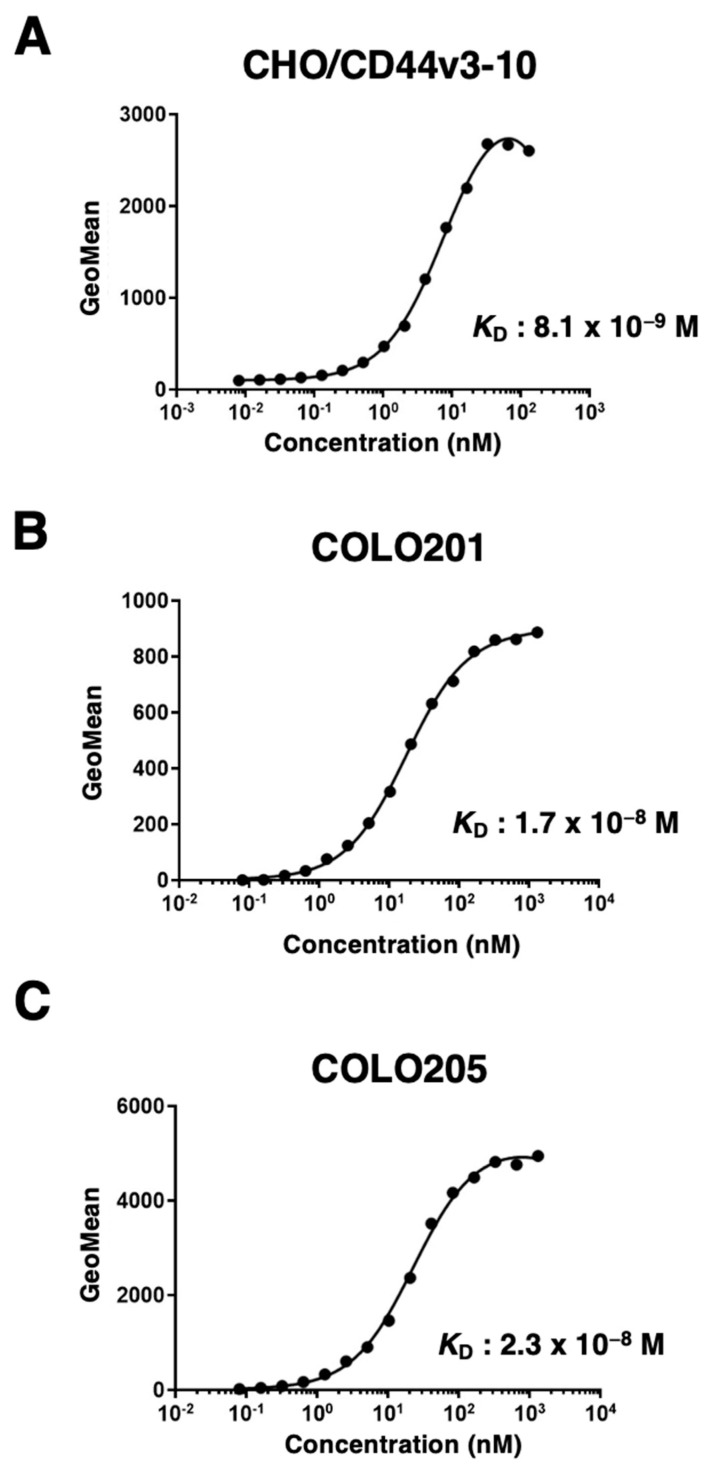
The determination of the binding affinity of C_44_Mab-9 to CD44-expressing cells. CHO/CD44v3-10 (**A**), COLO201 (**B**), and COLO205 (**C**) cells were suspended in 100 µL of serially diluted C_44_Mab-9 at indicated concentrations. Then, cells were treated with Alexa Fluor 488-conjugated anti-mouse IgG. Fluorescence data were subsequently collected, followed by the calculation of the apparent dissociation constant (*K*_D_) by GraphPad PRISM 8.

**Figure 4 ijms-24-04007-f004:**
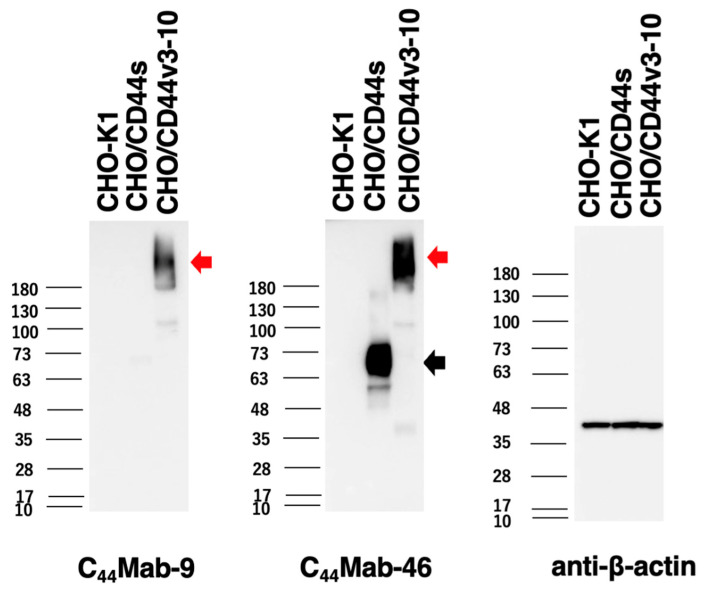
Western blotting by C_44_Mab-9. The cell lysates of CHO-K1, CHO/CD44s, and CHO/CD44v3-10 (10 µg of protein) were electrophoresed and transferred onto polyvinylidene difluoride (PVDF) membranes. The membranes were incubated with 10 µg/mL of C_44_Mab-9, 10 µg/mL of C_44_Mab-46, and 1 µg/mL of anti-β-actin mAb, followed by incubation with peroxidase-conjugated anti-mouse immunoglobulins. The black arrow indicates the CD44s (~75 kDa). The red arrows indicate the CD44v3-10 (>180 kDa).

**Figure 5 ijms-24-04007-f005:**
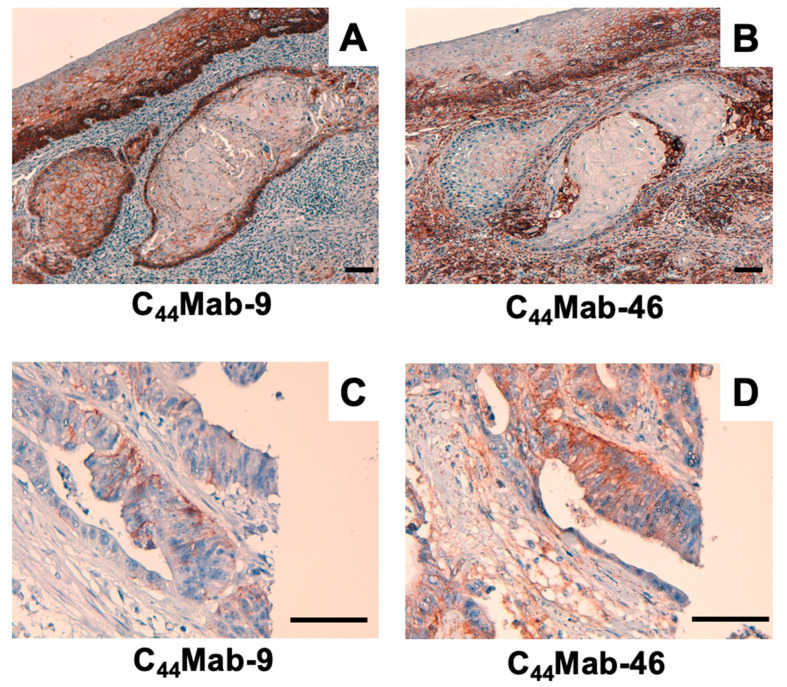
Immunohistochemical analysis using C_44_Mab-9 and C_44_Mab-46. (**A**,**B**) Oral SCC sections were incubated with 1 µg/mL of C_44_Mab-9 (**A**) and C_44_Mab-46 (**B**). (**C**,**D**) CRC sections were incubated with 1 µg/mL of C_44_Mab-9 (**C**) and C_44_Mab-46 (**D**), followed by treatment with the Envision+ kit. The color was developed using DAB, and sections were counterstained with hematoxylin. Scale bar = 100 µm.

**Table 1 ijms-24-04007-t001:** The determination of the binding epitope of C_44_Mab-9 by ELISA.

Peptide	Coding Exon *	Sequence	C_44_Mab-9
CD44p21–40	2	QIDLNITCRFAGVFHVEKNG	−
CD44p31–50	2	AGVFHVEKNGRYSISRTEAA	−
CD44p41–60	2	RYSISRTEAADLCKAFNSTL	−
CD44p51–70	2	DLCKAFNSTLPTMAQMEKAL	−
CD44p61–80	2/3	PTMAQMEKALSIGFETCRYG	−
CD44p71–90	2/3	SIGFETCRYGFIEGHVVIPR	−
CD44p81–100	3	FIEGHVVIPRIHPNSICAAN	−
CD44p91–110	3	IHPNSICAANNTGVYILTSN	−
CD44p101–120	3	NTGVYILTSNTSQYDTYCFN	−
CD44p111–130	3/4	TSQYDTYCFNASAPPEEDCT	−
CD44p121–140	3/4	ASAPPEEDCTSVTDLPNAFD	−
CD44p131–150	4/5	SVTDLPNAFDGPITITIVNR	−
CD44p141–160	4/5	GPITITIVNRDGTRYVQKGE	−
CD44p151–170	5	DGTRYVQKGEYRTNPEDIYP	−
CD44p161–180	5	YRTNPEDIYPSNPTDDDVSS	−
CD44p171–190	5	SNPTDDDVSSGSSSERSSTS	−
CD44p181–200	5	GSSSERSSTSGGYIFYTFST	−
CD44p191–210	5	GGYIFYTFSTVHPIPDEDSP	−
CD44p201–220	5	VHPIPDEDSPWITDSTDRIP	−
CD44p211–230	5/v3	WITDSTDRIPATSTSSNTIS	−
CD44p221–240	5/v3	ATSTSSNTISAGWEPNEENE	−
CD44p231–250	v3	AGWEPNEENEDERDRHLSFS	−
CD44p241–260	v3	DERDRHLSFSGSGIDDDEDF	−
CD44p251–270	v3/v4	GSGIDDDEDFISSTISTTPR	−
CD44p261–280	v3/v4	ISSTISTTPRAFDHTKQNQD	−
CD44p271–290	v4	AFDHTKQNQDWTQWNPSHSN	−
CD44p281–300	v4	WTQWNPSHSNPEVLLQTTTR	−
CD44p291–310	v4/v5	PEVLLQTTTRMTDVDRNGTT	−
CD44p301–320	v4/v5	MTDVDRNGTTAYEGNWNPEA	−
CD44p311–330	v5	AYEGNWNPEAHPPLIHHEHH	−
CD44p321–340	v5	HPPLIHHEHHEEEETPHSTS	−
CD44p331–350	v5/v6	EEEETPHSTSTIQATPSSTT	−
CD44p341–360	v5/v6	TIQATPSSTTEETATQKEQW	−
CD44p351–370	v6	EETATQKEQWFGNRWHEGYR	+
CD44p361–380	v6	FGNRWHEGYRQTPREDSHST	−
CD44p371–390	v6/v7	QTPREDSHSTTGTAAASAHT	−
CD44p381–400	v6/v7	TGTAAASAHTSHPMQGRTTP	−
CD44p391–410	v7	SHPMQGRTTPSPEDSSWTDF	−
CD44p401–420	v7	SPEDSSWTDFFNPISHPMGR	−
CD44p411–430	v7/v8	FNPISHPMGRGHQAGRRMDM	−
CD44p421–440	v7/v8	GHQAGRRMDMDSSHSTTLQP	−
CD44p431–450	v8	DSSHSTTLQPTANPNTGLVE	−
CD44p441–460	v8	TANPNTGLVEDLDRTGPLSM	−
CD44p451–470	v8/v9	DLDRTGPLSMTTQQSNSQSF	−
CD44p461–480	v8/v9	TTQQSNSQSFSTSHEGLEED	−
CD44p471–490	v9	STSHEGLEEDKDHPTTSTLT	−
CD44p481–500	v9/v10	KDHPTTSTLTSSNRNDVTGG	−
CD44p491–510	v9/v10	SSNRNDVTGGRRDPNHSEGS	−
CD44p501–520	v10	RRDPNHSEGSTTLLEGYTSH	−
CD44p511–530	v10	TTLLEGYTSHYPHTKESRTF	−
CD44p521–540	v10	YPHTKESRTFIPVTSAKTGS	−
CD44p531–550	v10	IPVTSAKTGSFGVTAVTVGD	−
CD44p541–560	v10	FGVTAVTVGDSNSNVNRSLS	−
CD44p551–570	v10/16	SNSNVNRSLSGDQDTFHPSG	−
CD44p561–580	v10/16	GDQDTFHPSGGSHTTHGSES	−
CD44p571–590	16/17	GSHTTHGSESDGHSHGSQEG	−
CD44p581–600	16/17	DGHSHGSQEGGANTTSGPIR	−
CD44p591–606	17	GANTTSGPIRTPQIPEAAAA	−

+, OD655 ≧ 0.3; −, OD655 < 0.1. * The CD44 exons are illustrated in Figure 1.

## Data Availability

All related data and methods are presented in this paper. Additional inquiries should be addressed to the corresponding authors.

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
