# Peer review of "Development of a Novel Anti-CD44 Variant 6 Monoclonal Antibody C44Mab-9 for Multiple Applications against Colorectal Carcinomas"

_ijms, 2023, doi:10.3390/ijms24044007_

Round 1

Reviewer 1 Report

The present study by Ejima R et al. provides an exciting overview of the role of CD44 in colorectal cancer and clearly describes the development and production of anti-CD44 monoclonal antibodies that were successfully used in clinical samples. I would recommend authors provide more information regarding the changes required to reform the produced anti-CD44 antibodies to be also used for therapeutic purposes (e.g. humanized antibodies, etc.)

Author Response

The present study by Ejima R et al. provides an exciting overview of the role of CD44 in colorectal cancer and clearly describes the development and production of anti-CD44 monoclonal antibodies that were successfully used in clinical samples. I would recommend authors provide more information regarding the changes required to reform the produced anti-CD44 antibodies to be also used for therapeutic purposes (e.g. humanized antibodies, etc.)

According to the comment, we added the discussion about the possibility of C44Mab-9 for CAR-T therapy (line 202-). We introduced the current application of CD44v6 CAR-T and its problem. Furthermore, we discussed the beneficial possibility of C44Mab-9.

Reviewer 2 Report

Authors developed a CD44 Mab and determined its epitope as the variant 6 encoded region, which plays critical roles in cell proliferation, migration, survival, and angiogenesis. They then characterized this Mab using multiple techniques, including flow cytometry, western blotting, and immunohistochemistry. Overall, the manuscript is clearly written, the methods well designed, and the experimental data generally supports the main conclusions of the paper. However, I have one concern that should be addressed prior to any publication.

  1. In the discussion, could the authors include mention on how this C44Mab-9 is different than the already developed VFF18 Mab for CD44V6? From the details provided it seems that the VFF18 Mab for CD44V6 serves essentially the same functions as the C44Mab-9 being developed in this study.

Author Response

Authors developed a CD44 Mab and determined its epitope as the variant 6 encoded region, which plays critical roles in cell proliferation, migration, survival, and angiogenesis. They then characterized this Mab using multiple techniques, including flow cytometry, western blotting, and immunohistochemistry. Overall, the manuscript is clearly written, the methods well designed, and the experimental data generally supports the main conclusions of the paper. However, I have one concern that should be addressed prior to any publication.

  1. In the discussion, could the authors include mention on how this C44Mab-9 is different than the already developed VFF18 Mab for CD44V6? From the details provided it seems that the VFF18 Mab for CD44V6 serves essentially the same functions as the C44Mab-9 being developed in this study.

According to the comment, we added the discussion about the possibility of C44Mab-9 for CAR-T therapy (line 202-). We introduced the current application of CD44v6 CAR-T (derived from VFF18) and its problem. Furthermore, we discussed the different points and beneficial possibility of C44Mab-9 compared to VFF18.

Reviewer 3 Report

The paper "Development of a Novel Anti-CD44 variant 6 Monoclonal Antibody for Multiple Applications against Colorectal Carcinomas" by Ryo Ejima et al., describes the development of a new, novel  anti-CD44 monoclonal antibody variant. It is clearly shown that the antibody exhibits unique binding in vitro.  However, relevant studies should have been done, not speculated in the discussion ( in vivo, organoids etc.) to demonstrate the superior use of this clone.  The authors speculate that afucosylation is needed. It can be useful if they could decipher the mechanism of action of this antibody.    The authors have the capacity to perform these experiments.

Also, the antibody sequence is expected.

I think this is not enough for publishing in this journal

Author Response

The paper "Development of a Novel Anti-CD44 variant 6 Monoclonal Antibody for Multiple Applications against Colorectal Carcinomas" by Ryo Ejima et al., describes the development of a new, novel anti-CD44 monoclonal antibody variant. It is clearly shown that the antibody exhibits unique binding in vitro. However, relevant studies should have been done, not speculated in the discussion (in vivo, organoids etc.) to demonstrate the superior use of this clone. The authors speculate that afucosylation is needed. It can be useful if they could decipher the mechanism of action of this antibody. The authors have the capacity to perform these experiments.

Also, the antibody sequence is expected.

I think this is not enough for publishing in this journal

As we mentioned in discussion, investigation of the biological activity is one of the important future studies.

Moreover, we would like to focus on the therapeutic application of C44Mab-9. As we added in the discussion, we mentioned the possibility of C44Mab-9 for CAR-T therapy (line 202-). We introduced the current application of CD44v6 CAR-T (derived from VFF18) and its problem. Furthermore, we discussed the beneficial possibility of C44Mab-9 compared to VFF18.

We have started the cloning of antibody sequence to develop the recombinant/afucosylated antibody and CAR for therapeutic uses.

We hope the reviewer to concider the future prospects of C44Mab-9.